# Building and identifying highly active oxygenated groups in carbon materials for oxygen reduction to $H_2O_2$

Gao-Feng Han [1], Feng Li [1✉], Wei Zou[2], Mohammadreza Karamad[3], Jong-Pil Jeon[1], Seong-Wook Kim[1], Seok-Jin Kim [1], Yunfei Bu [4], Zhengping Fu[2,5], Yalin Lu [2,5], Samira Siahrostami [6✉] & Jong-Beom Baek [1✉]

The one-step electrochemical synthesis of $H_2O_2$ is an on-site method that reduces dependence on the energy-intensive anthraquinone process. Oxidized carbon materials have proven to be promising catalysts due to their low cost and facile synthetic procedures. However, the nature of the active sites is still controversial, and direct experimental evidence is presently lacking. Here, we activate a carbon material with dangling edge sites and then decorate them with targeted functional groups. We show that quinone-enriched samples exhibit high selectivity and activity with a $H_2O_2$ yield ratio of up to 97.8 % at 0.75 V vs. RHE. Using density functional theory calculations, we identify the activity trends of different possible quinone functional groups in the edge and basal plane of the carbon nanostructure and determine the most active motif. Our findings provide guidelines for designing carbon-based catalysts, which have simultaneous high selectivity and activity for $H_2O_2$ synthesis.

[1] School of Energy and Chemical Engineering/Center for Dimension-Controllable Organic Frameworks, Ulsan National Institute of Science and Technology (UNIST), Ulsan 44919, South Korea. [2] CAS Key Laboratory of Materials for Energy Conversion, Department of Materials Science and Engineering, University of Science and Technology of China (USTC), 230026 Hefei, P. R. China. [3] Department of Chemical and Petroleum Engineering, University of Calgary, 2500 University Drive NW, Calgary, AB T2N 1N4, Canada. [4] Jiangsu Key Laboratory of Atmospheric Environment Monitoring and Pollution Control, School of Environmental Science and Engineering, Nanjing University of Information Science and Technology (NUIST), 219 Ningliu, 210044 Nanjing, Jiangsu, P. R. China. [5] Synergetic Innovation Center of Quantum Information and Quantum Physics and Hefei National Laboratory for Physical Sciences at Microscale, University of Science and Technology of China (USTC), 230026 Hefei, P. R. China. [6] Department of Chemistry, University of Calgary, 2500 University Drive NW, Calgary, AB T2N 1N4, Canada. ✉email: lifeng@unist.ac.kr; samira.siahrostami@ucalgary.ca; jbbaek@unist.ac.kr

Hydrogen peroxide is widely used as a green oxidant in disinfectants, bleaching agents, sanitizing agents, chemical synthesis, and even as a potential energy carrier[1–14]. According to a report by Global Industry Analysts, Inc.[15], the global consumption of $H_2O_2$ is projected to reach 6.0 million metric tons by 2024. Approximately 99% of all $H_2O_2$ is currently produced using the multi-step anthraquinone process. However, this process is energy intensive, and can only be performed in a centralized factory[1,5,10]. The inherent complexity of the anthraquinone process has driven many researchers to investigate one-step methods, which can continuously produce $H_2O_2$ on-site at small scale using a simple device[1,5]. Among these, the electrochemical synthesis approach is one of the best choices to meet the above demands.

Metal-free oxidized carbon materials have received particular attention for this application because they offer a simple synthetic procedure, cost-effectiveness, as well as high activity and selectivity[16–29]. They allow the incorporation of a variety of oxygen functional groups, with the potential to widely tune performance, and to optimize active sites. For example, Lu et al. recently used nitric acid to oxidize carbon nanotubes and demonstrated that the resulting oxidized carbon-material was highly active for oxygen reduction to hydrogen peroxide (ORHP)[16]. They considered the active sites to be the carbon atoms adjacent to carboxylic acid and etheric groups (–COOH and C–O–C). Kim et al. used few-layer mildly reduced graphene oxide electrodes, by partially removing oxygen from graphene oxide using hydrothermal heating without a reducing agent, and showed that $sp^2$-hybridized carbon near ring ethers along the sheet edges were the most active sites for ORHP[17].

All of these pioneering studies apply harsh oxidation conditions to cleave the strong $sp^2$ C–C bonds[16,30] and incorporate oxygen functional groups, and this in turn makes it difficult to functionalize the carbon materials with targeted groups. As a result, the oxygen functional groups inevitably saturate the dangling bonds on the edges with sophisticated multi-components, which hinders systematic experimental study. It becomes inherently difficult to accurately identify the most active functional groups, because the resolution of ordinary characterization methods is not high enough to distinguish between similar groups.

From the industrial anthraquinone process, we know that quinones are the champion catalytically active oxygen functional groups towards $H_2O_2$ synthesis[10,31]. Herein we demonstrate a facile synthesis method to incorporate quinone functional groups in carbon nanostructures. For comparison, we also employ a pre-activated method to build carbonyl-enriched and etheric ring-enriched graphitic nanoplatelets (denoted as $GNP_{C=O}$ and $GNP_{C–O–C}$, respectively). Using fine characterization methods, such as soft XANES, XPS, FTIR, and CV, we show that each of our samples have targeted and desired functional groups (etheric ring, carboxyl and quinone). Electrochemical measurements reveal that the sample with abundant quinone functional group ($GNP_{C=O,1}$) exhibits high selectivity, with a $H_2O_2$ yield ratio of 97.8% at 0.75 V, which is superior to previously reported etheric ring and carboxylic acid groups.

We further demonstrate the activity of the quinone functional groups by examining standalone quinone molecular systems. Finally, we use density functional theory (DFT) calculations to model different possible quinone functional groups and examine their activity toward ORHP. We find that the quinone groups exhibit very high activity with negligible overpotential.

## Results

**Synthesis and structure characterization**. We adopted a two-step method to prepare edge-oxygenated graphitic nanoplatelets.

The free edge sites were first created by simultaneously crushing and exfoliating graphite mechanochemically. Then, the reactive edge sites were saturated with mild oxidants of $CO_2$ or diluted $O_2$. We found that when $CO_2$ was applied as an oxidant, carbonyl-related groups were easily formed. In this case, the as-prepared product is designated $GNP_{C=O}$. Even dilute $O_2$ could also be selected as an oxidant. Because the electronic ground state of $O_2$ and carbon are different, the triplet $O_2$ cannot react with singlet defect-free graphitic carbon spontaneously, due to the energy barrier present. This allows selective oxidation of the reactive edge sites with a high degree of controllability. Etheric ring (C–O–C)-functionalized graphitic nanoplatelets ($GNP_{C–O–C}$) were obtained in dilute $O_2$ atmosphere.

We first characterized the morphology of the above samples using field emission scanning electron microscopy (FESEM, Supplementary Fig. 1) and transmission electron microscopy (TEM, Supplementary Fig. 2). This analysis showed that $GNP_{C=O}$ and $GNP_{C–O–C}$ had morphologies typical of graphitic nanoplatelets and nanoparticles, respectively. The successful nanosizing and functionalization were further verified by Raman spectra and X-ray powder diffraction (XRD) patterns. As shown in Supplementary Fig. 3, the strong D band (usually as an indicator of defective edges) and broad (002) facets (derived according to the Scherrer equation), together indicate the existence of plenty of edge sites in the nanosized GNPs. The Brunauer–Emmett–Teller (BET, Supplementary Fig. 4) analysis determined that the specific surface areas of $GNP_{C=O,1}$, $GNP_{C=O,2}$, and $GNP_{C–O–C}$, are 450, 753, and 657 $m^2 g^{-1}$, respectively.

Next, we measured sample oxygen content using three different methods, elemental analysis (EA), energy dispersive spectroscopy (EDS), and XPS. The results (summarized in Supplementary Table 1 and Figs. 5–7) agreed with each other well. Here, we highlight the EA results because it has higher accuracy than the other two methods. The amount of oxygen content was calculated to be 10.5 at% for $GNP_{C=O,1}$, 20.6 at% for $GNP_{C=O,2}$, and 10.0 at% for $GNP_{C–O–C}$. A possible iron remnant was checked by time-of-flight secondary ion mass spectrometry (TOF-SIMS), which has a detection limit as low as ppm. The results (Supplementary Fig. 6) showed that there was no detectable iron in the samples.

Finally, to carefully unravel the nature of the oxygen functional groups we used a combination of techniques, including soft XANES, XPS, FTIR, and CV. Among them, soft XANES is one of the most powerful tools for characterizing graphitic materials, and can provide important information about bonding configurations with high resolution[32–34].

As shown in Fig. 1a, the C K-edge includes unoccupied $\pi^*$ (peak A–C) and excited $\sigma^*$ (peak D, E) states. Here, we only discuss the fingerprint region of $\pi^*$ in detail. The peak A around 285.4 eV is assigned to $1s – \pi^*$ from $sp^2$ C=C[32–36]. The peak B results from the charge transfer induced by the O in the etheric ring. The $GNP_{C–O–C}$ and partially reduced graphene oxide ($p$RGO, Supplementary Methods) exhibit a minor shoulder at 287.2 eV (Peak $B_1$), which corresponds to the out-of-plane etheric ring (C–O–C)[32,34,35]. Since the in-plane C–O bonding of C–O–C is much stronger than the out-of-plane, the peak at 288.2 eV is supposed to be in-plane C–O–C at the edge of the GNP. Thus, the $GNP_{C–O–C}$ is mainly composed of in-plane C–O–C, and partly by out-of-plane C–O–C. In $p$RGO there is only out-of-plane C–O–C. The peak C around 288.6 eV is attributed to $\pi^*$(C=O)[32,35], which should be determined in combination with other characterizations.

The soft O K-edge XANES (Fig. 2b) provides further information about the groups. Here, the $p$RGO shows an exclusive $A_1$ peak (530.1 eV), which is assigned to the $\pi^*$(C=O) of the organic carbonate. The peak $A_2$ (531.0 eV) is assigned to quinone contributions[33]. The results of the O K-edge demonstrate that the $GNP_{C=O,1}$ is enriched with quinone. The presence

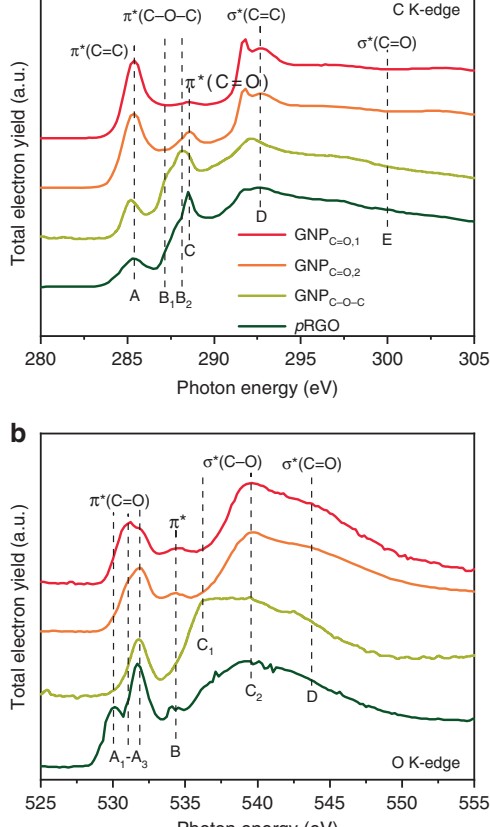

**Fig. 1 The soft X-ray absorption near-edge structure (XANES). a** C K-edge. A, 1s – π* from $sp^2$ C=C; B, 1s – π* from the etheric ring, B₁, the out-of-plane C-O-C, B₂, the in-plane C-O-C; C, 1s – π* from ketone or carboxylic acid; D, 1s – σ* from $sp^2$ C=C; D, 1s - σ* from C=O. **b** O K-edge. A, 1s – π* from C=O, A₁, organic carbonate, A₂, quinone, A₃, ketone or carboxylic acid; B, the 1s – π* excitations due to the charge transfer between C and O, including C=O and C-O; C, 1s - σ* from C-O, C₁, etheric ring (C-O-C), C₂, the C-O in COOH; C, 1s - σ* from C=O. The soft XANES were collected in the total electron yield (TEY) mode.

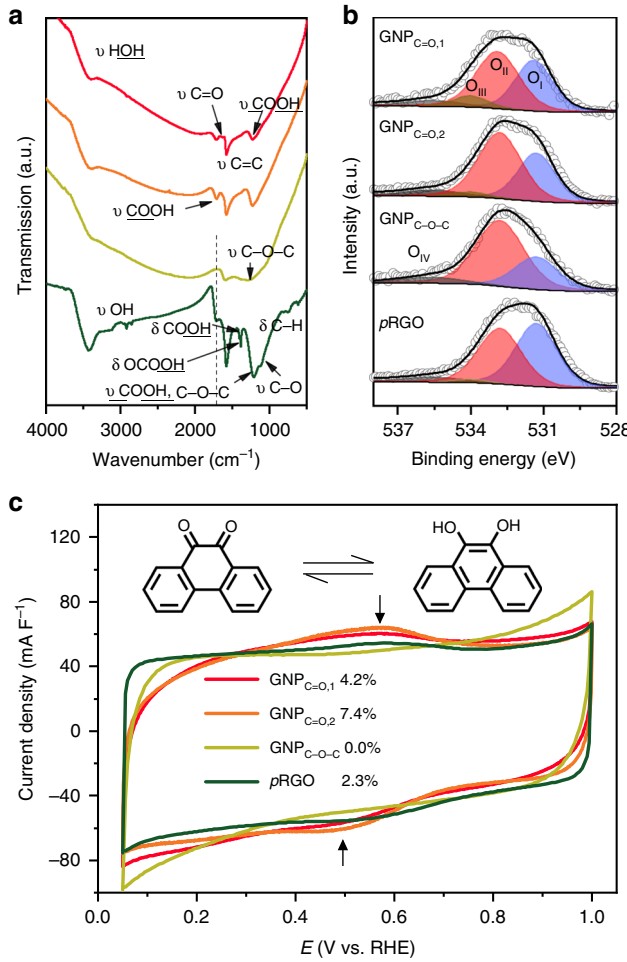

**Fig. 2 Edge group characterization. a** Fourier transform infrared (FTIR) spectra. **b** The high-resolution O 1s of the X-ray photoelectron spectra (XPS). O₁, 531.35 ± 0.05 eV, C=O, quinone, or ketone; O₁₁, 532.8 eV, C-O-C, or COOH; O₁₁₁, 534.0 eV, C-O(H); O₁ᵥ, 535.6 eV, adsorbed $H_2O$ and $O_2$. **c** The cyclic voltammetry. The CV curves were measured in Ar-saturated 0.5 M $H_2SO_4$ at a scan rate of 50 mV s⁻¹. The quinone redox reaction were recorded in the potential window from 0.3 to 0.8 V.

of a peak A₂ shoulder indicates the GNP$_{C=O,2}$ and *p*RGO also contain less quinone groups. Peak A₃ (531.8 eV) may be composed of ketones or/and carboxylic acid[32–34]. Fortunately, peak A₃ can be identified by peak C₂ (539.6 eV), which originates from the σ*(C-O) in carboxylic acid (COOH)[32–34,36]. Peaks A₃ and C₂ are dominant in GNP$_{C=O,2}$ and *p*RGO, which indicates that they are mainly composed of COOH.

We can draw the conclusion that the quinone was easily formed in the short reaction time (GNP$_{C=O,1}$). However, carboxylic acid was the main group after the prolonged reaction time (GNP$_{C=O,2}$). The contribution of π* peak B is complicated. It originates from the resonance transition of quinone, or the charge transfer due to the influence of the neighboring O in the etheric ring[33]. The GNP$_{C-O-C}$ gives rise to a σ* resonance at 536.3 eV (peak C₁), which is attributed to C-O-C[34]. Here, we noticed that GNP$_{C-O-C}$ contains peak A₃, and a small shoulder of π*(C=O) in the C K-edge, which indicates that some carbonyl-related groups are present in the GNP$_{C-O-C}$.

FTIR is another powerful tool for characterizing functional groups, and was employed here. As shown in Fig. 2a, all of the spectra exhibit a broad band at around 3400 cm⁻¹, which is mainly assigned to the stretching vibration (υ OH) of the adsorbed moisture in KBr. Another common band, located at 1610 cm⁻¹, is attributed to the asymmetric stretching of $sp^2$-hybridized C-C

bonds (υ C=C). Its shifting and intensity are intimately related to the oxygenated groups[37,38].

The fingerprint region, ranging from 1750 to 1000 cm⁻¹, was scrutinized to identify the functional groups present. The well-documented *p*RGO was analyzed first. Consistent with other reports[38–42], the functional groups in the *p*RGO were assigned to ketones (υ C=O, the shoulder at 1654 cm⁻¹), carboxylic acid (υ C=O at 1715 cm⁻¹, δ OH at 1402 cm⁻¹, υ C-OH at 1214 cm⁻¹), organic carbonate (υ C=O at 1715 cm⁻¹, δ OH at 1384 cm⁻¹), ethers (υ C-O at 1084 cm⁻¹), and aromatic hydrocarbons (δ C-H at 740 cm⁻¹)[38–42].

Both GNP$_{C=O,1}$ and GNP$_{C=O,2}$ produced a typical fingerprint band of carboxylic acid (υ C=O in COOH, 1716 cm⁻¹). The COOH is formed via hydrolysis in acid[43,44]. GNP$_{C=O,1}$ and GNP$_{C=O,2}$ also displayed a peak and a shoulder, respectively, at 1632 cm⁻¹, which was thought to be the quinone based on the XANES results. The GNP$_{C-O-C}$ displayed a broad band from 1380 to1050 cm⁻¹, which was attributed to the asymmetric C-O-C stretching vibration (in-plane υ C-O-C) of the etheric rings. This is caused by an unusual absorption mechanism[40].

The XPS was then performed (Supplementary Fig. 7). It is known that that the C 1s typically exhibits a pronounced

asymmetric tail at higher binding energy[45], making it difficult to obtain accurate deconvolution results, especially considering the controversial assignment. Here, we only focused on the O 1s spectroscopy.

The high-resolution O 1s are shown in Fig. 2b. After careful deconvolution with the same standard rules, four distinct regions were identified. To easily distinguish them, we labelled them $O_I$ (531.35 ± 0.05 eV, C=O related groups)[46,47], $O_{II}$ (532.8 eV, C–O–C or COOH)[46,47], $O_{III}$ (534.0 eV, C–O(H))[46], and $O_{IV}$ (535.6 eV, physically adsorbed $H_2O$ and $O_2$)[41,42]. Because the different groups overlap, it was difficult to determine the contents of the groups by O 1s alone. However, we can still draw some useful information. All samples had a very small $O_{III}$ region, which indicates that very little phenolic -OH is present. Based on the XANES and FTIR results, it was determined that the $O_{II}$ regions in $GNP_{C=O,1}$, $GNP_{C=O,2}$, and $pRGO$ mainly result from COOH. However, the $O_{II}$ region in $GNP_{C-O-C}$ originates from C–O-C. $GNP_{C=O,1}$ had a higher percentage of $O_I$ relative to $GNP_{C=O,2}$ due to the higher content of quinone. The $GNP_{C-O-C}$ exhibited the smallest $O_I$ region. In contrast, the $pRGO$ exhibited the highest $O_I$ region. However, the ketone and quinone in the $O_I$ region still could not be distinguished.

Fortunately, quinone is sensitive to cyclic voltammetry (CV). Since the quinone redox couple is the only reaction that can be detected by CV, we used CV to characterize the content of quinone in the samples (Supplementary Fig. 8)[10,31]. As shown in Fig. 2c, the quinone content was 4.2%, 7.4%, 0.0%, and 2.3% for $GNP_{C=O,1}$, $GNP_{C=O,2}$, $GNP_{C-O-C}$, and $pRGO$, respectively. The absence of quinone redox indicates that the $O_I$ region in $GNP_{C-O-C}$ is attributed to the presence of ketone. The reduced quinone redox content in $pRGO$ demonstrates that most of the $O_I$ region is composed of COOH. The functional groups of the samples are summarized in Supplementary Table 2.

**Oxygen reduction to hydrogen peroxide**. The oxygen reduction to hydrogen peroxide (ORHP) performance of the synthesized samples with different oxygen groups was evaluated using a rotating ring-disk electrode (RRDE). Since $H_2O_2$ can upshift the potential of the reference electrode because of its strong oxidizing nature (Supplementary Fig. 9), a salt bridge must be used to separate the electrochemical cell and reference electrode (Supplementary Fig. 10). For a fair comparison of activity, all of the tested samples were tuned to the same capacitance by changing the volume of drop-cast ink. We used a reversible hydrogen electrode (RHE, Supplementary Fig. 11) for all our analyses.

The polarization curves were measured in $O_2$-saturated 0.1 M aq. KOH solution at a scan rate of 1600 rpm. To eliminate the contributions of capacitance, we averaged the current of the forward and backward scans. The $H_2O_2$ current ($J_{H2O2}$) was collected at the ring electrode ($J_R$) with an applied potential of 1.15 V, and the collection efficiency was 37% (Supplementary Fig. 12). The four-electron byproducts of $H_2O$ were calculated using the relationship: $J_{H2O} = J_D − J_R$.

As shown in Fig. 3a and Supplementary Table 3, the carbonyl-enriched samples, $GNP_{C=O,1}$ (0.826 V), $GNP_{C=O,2}$ (0.815 V), and $pRGO$ (0.810 V), exhibited a higher onset potential than $GNP_{C-O-C}$ (0.805 V). Here, the onset potential is defined as the potential measured at a current density of 0.15 mA cm$^{-2}$ (5% of the theoretical limiting current) for ORHP. The result indicates that the sample with the in-plane etheric ring had lower activity than the carbonyl-enriched samples.

It is noteworthy that all of the onset potentials were even higher than the thermodynamic equilibrium potential ($O_2 + H_2O + 2 e$ ⇌ $HO_2^− + OH^−$, 0.75 V vs RHE). This counterintuitive phenomenon has been widely reported[16,48]. Bao and his colleagues

attributed this to a Nernst-related potential shift due to the low concentration of $H_2O_2$ in the electrolyte, and/or a possible pH-related change[48]. Although $H_2O_2$ can upshift the potential of the reference electrode, its influence can be easily ruled out by employing a salt bridge and refreshing the electrolyte for every measurement. This abnormal phenomenon did not exist with a planar electrode, which further confirms that the potential shifting is unrelated to the yield of $H_2O_2$. We suggest this abnormal phenomenon only results from the localized pH-related change in the mesoporous electrode due to the constraint of mass transport.

The kinetic current of $H_2O_2$ ($J_{K,H2O2}$) was calculated according to the Koutecky-Levich equation: $1/J_{H2O2} = 1/J_{K,H2O2} + 1/J_{L,H2O2}$, where $J_{K,H2O2}$ is the measured current of the $H_2O_2$ yield, $J_{L,H2O2}$ is the theoretical limiting current of ORHP[23,49]. $J_{L,H2O2}$ was obtained from the Levich equation and determined to be 2.9 mA cm$^{-2}$. The calculated results are shown in the Tafel plot in Fig. 3b. $GNP_{C=O,1}$, and $GNP_{C=O,2}$ showed the smallest Tafel slope (48 mV dec$^{-1}$), which was superior to the C–O–C-enriched $GNP_{C-O-C}$ (55 mV dec$^{-1}$). The same Tafel slope indicates similar active sites. Although the $pRGO$ was also carbonyl-enriched, its Tafel slope was only 63 mV dec$^{-1}$. This is possibly due to the considerable N content, which is inevitably introduced during the preparation of $pRGO$. The doped N is known to enhance the yield of 4e product $H_2O$[50].

The $J_{K,H2O2}$ at 0.65 and 0.75 V were selected to compare activity (Fig. 3c). We note that $GNP_{C=O,1}$ had the highest activity (25.1 mA cm$^{-2}$) at 0.65 V. However, the most active sample was $GNP_{C=O,2}$ (3.6 mA cm$^{-2}$) at a higher potential of 0.75 V. This phenomenon is caused by the samples′ different selectivity (the $H_2O_2$ yield ratio, Fig. 3d), which plays a major role in the low potential region. The $GNP_{C=O,1}$ exhibited higher selectivity (the $H_2O_2$ yield ratio, 97.8% at 0.75 V) with an electron transfer number of nearly 2 (Supplementary Fig. 13), which is the highest ratio reported so far (Supplementary Table 3)[16,18,19,21,22,25–29,48]. The $pRGO$ exhibited the poorest $J_{K,H2O2}$ (1.2 mA cm$^{-2}$ at 0.75 V) and the lowest $H_2O_2$ yield ratio (83.4% at 0.75 V) because of the presence of graphitic N (Supplementary Fig. 14), which is reported to be the active sites for 4e ORR[50].

The $H_2O_2$ yield ratio of COOH-enriched $GNP_{C=O,2}$ decreased relative to the quinone-enriched $GNP_{C=O,1}$. The possible reason is that COOH has a negative impact on selectivity. The $H_2O_2$ yield ratio of all of the carbonyl-enriched samples decreased as we lowered the potential. This may have occurred due to the reduction of quinone to hydroquinone at low potential. Distinctively, the $GNP_{C-O-C}$ displayed a tendency opposite to the others, due to the absence of quinone.

The ORHP performance was also evaluated in neutral (0.05 M $Na_2SO_4$) and acidic (0.1 M $HClO_4$) media. The results are shown in Supplementary Fig. 15. Both results in neutral and acidic media demonstrated that the quinone-enriched sample ($GNP_{C=O,1}$) had the highest selectivities, which were 95% (0.6 V in neutral) and 85% (0.2 V in acid), respectively. In the neutral medium, the onset potential is the same with the thermodynamic equilibrium potential (0.70 V), which indicates the absence of potential shift, which is caused by the localized pH-related change. In the acidic medium, the activity toward ORHP is poor. The result agrees well with the previous reports[16–18].

Besides selectivity and activity, stability is also one of the three indispensable factors for catalysts, and it affects their economical durability. The samples′ stability was measured in an H-type cell, which was separated by a Nafion 115 membrane (Supplementary Fig. 16). It is known that the alkaline $H_2O_2$ solution is unstable, and spontaneous decomposition will occur[1,51,52]. Approximately 30% of $H_2O_2$ was decomposed during a 30 h stability test (Supplementary Fig. 17). Therefore, $MgSO_4$, a popular stabilizer in alkaline solution, was selected to suppress the decomposition[51].

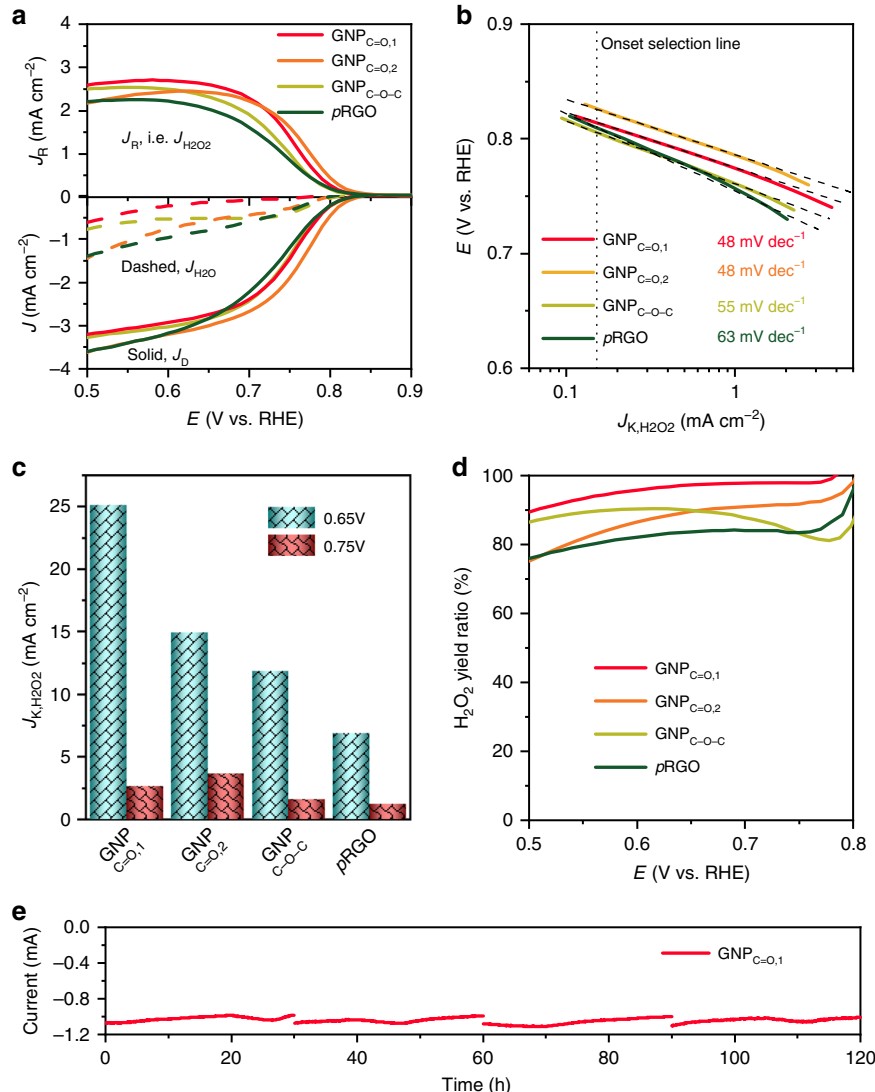

**Fig. 3 The performance characterizations of ORHP. a** The polarization curves of $H_2O_2$ current ($J_R$ or $J_{H2O2}$), disk current ($J_D$), and $H_2O$ current ($J_{H2O}$). The curves were measured in $O_2$-saturated 0.1 M KOH solution at a scan rate of 10 mV s$^{-1}$ by RRDE with a rotation speed of 1600 rpm. The applied potential of the ring was 1.15 V. The current was the average of the forward and backward scans. **b** The corresponding Tafel plot of kinetic current of $H_2O_2$ ($J_{K,H2O2}$). **c** The $J_{K,H2O2}$ comparison at 0.65 and 0.75 V. **d** The corresponding $H_2O_2$ yield ratio. **e** Stability tests in four fresh 0.1 M KOH electrolytes, which were measured in a H-type cell, with 400 ppm MgSO$_4$ as stabilizer to suppress the decomposition of $H_2O_2$. The applied potential was 0.65 V.

As shown in Fig. 3e, the current exhibited a slow decline as the time increased. The possible reason is that the increasing $H_2O_2$ concentration enhances electrolyte viscosity. The increasingly sluggish diffusion of $O_2$ in the $H_2O_2$ solution subsequently deteriorates the ORHP current[53]. This speculation was further verified by the observation that there was no current drop after changing to a fresh electrolyte. Even after 120 h, there was no evident decline, which demonstrates that the catalyst has good stability.

The concentration of generated $H_2O_2$ was determined using the $KMnO_4$ titration method (Supplementary Movie 1)[28]. With an applied potential of 0.65 V in 90 mL electrolyte, a typical concentration of $H_2O_2$ after 30 h reaction is about 6.1 mM, with a Faraday efficiency[11,23] of 95%.

**Determination of active sites.** So far, we have shown that the carbonyl-enriched groups possess higher activity than the etheric ring groups. However, the nature of the active carbonyl groups remains elusive. To elaborate the nature of the active carbonyl groups, we designed a special experiment.

We tuned the content of quinone and carboxylic acid by leaching the sample of GNP$_{C=O,2}$ in an acidified concentrated $H_2O_2$ solution. The leached sample was labelled GNP$_{C=O,3}$. As shown in Fig. 4a, FTIR determined that the carboxylic acid content evidently increased after leaching. However, the quinone content was reduced from 7.4% to 4.5% (Fig. 4b). This change in groups was also recorded by XPS (Supplementary Fig. 18). The active sites can be easily identified by comparing the ORHP performance of GNP$_{C=O,2}$ and GNP$_{C=O,3}$.

The polarization curves are shown in Fig. 4c. The GNP$_{C=O,3}$ had both a lower onset potential of 14 mV and $J_{K,H2O2}$ (3.1 mA cm$^{-2}$, Supplementary Fig. 19) than GNP$_{C=O,2}$. The relation of $J_{K,H2O2}$ and quinone content was further checked, and the results are shown in Fig. 4d. $J_{K,H2O2}$ increased as the quinone content increased. Because of the influence of other groups, it did not follow a linear relation. These results confirmed that the active sites were quinone.

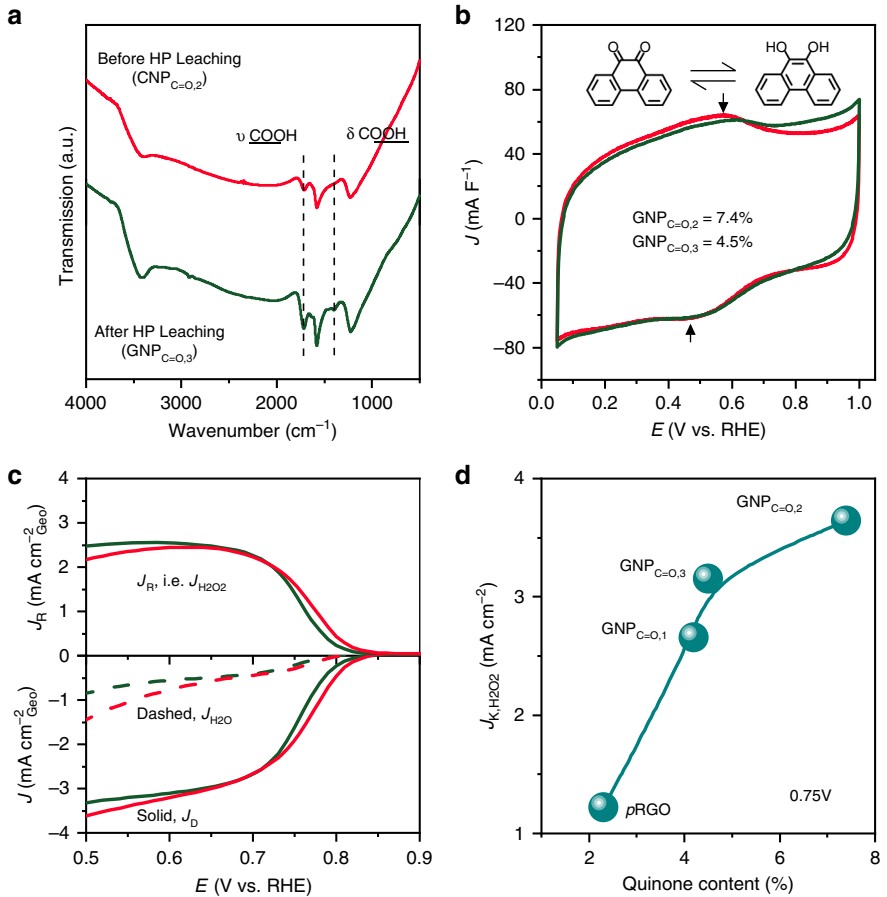

**Fig. 4 Determination of active sites. a** FTIR. The COOH is increased. **b** Cyclic voltammetry, which was measured in Ar-saturated 0.5 M $H_2SO_4$ at a scan rate of 50 mV s$^{-1}$. The quinone is decreased. **c** The polarization curves of $J_{H2O2}$, $J_D$, and $J_{H2O}$. The curves were measured in $O_2$-saturated 0.1 M KOH solution at a scan rate of 10 mV s$^{-1}$ by RRDE with a rotation speed of 1600 rpm. The current was the average of the forward and backward scan. **d** The $J_{K, H2O2}$ as a function of quinone content.

To further verify these results, we investigated several standalone molecules with quinone, carboxylic acid, and etheric ring groups, such as phenanthrenequinone, anthraquinone, naphthalenetetracarboxylic dianhydride, perylenetetracarboxylic dianhydride, dibenzodioxin, and dibenzofuran. The polarization curves are shown in Fig. 5. Except for phenanthrenequinone and anthraquinone, the other four molecules did not show activity towards ORHP; the activity was inferior to blank glass carbon (GC). The phenanthrenequinone was superior to anthraquinone in both the $J_{K,H2O2}$ (0.7 vs 0.5 mA cm$^{-2}$ at 0.65 V) and Tafel slope results (45 vs 48 mV dec$^{-1}$, Supplementary Fig. 20). These molecular chemistry results further confirm that the quinones are the active sites.

**Theoretical investigation**. To gain atomistic insights about the nature of the active quinone motifs, we next used density functional theory (DFT) calculations. We examined a variety of model structures (Fig. 6a) to study the different possible quinone groups on the edge and basal planes. These model structures were used to model the ORHP reaction pathway (Eqs. 1 and 2)[5,16]:

$$O_2 + H_2O + e^- - + * \rightarrow OOH* + OH^- - \qquad (1)$$

$$OOH* + e^- - \rightarrow HO_2^- + * \qquad (2)$$

where the $O_2$ molecule adsorbs at the carbon surface and is reduced through the first proton-electron transfer to form OOH*

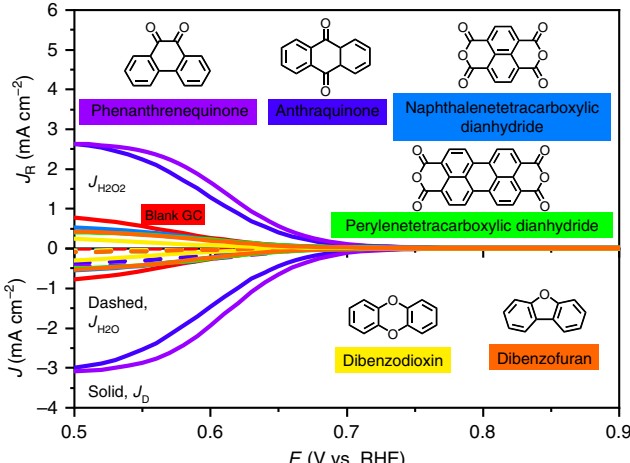

**Fig. 5 The ORHP performance of standalone molecules.** Phenanthrenequinone, anthraquinone, naphthalenetetracarboxylic dianhydride, perylenetetracarboxylic dianhydride, dibenzodioxin, dibenzofuran, and blank glass carbon (GC) were compared. The curves were measured in $O_2$-saturated 0.1 M KOH solution at a scan rate of 10 mV s$^{-1}$ by RRDE with a rotation speed of 1600 rpm. The applied potential of the ring was 1.15 V. The current was the average of the forward and backward scans.

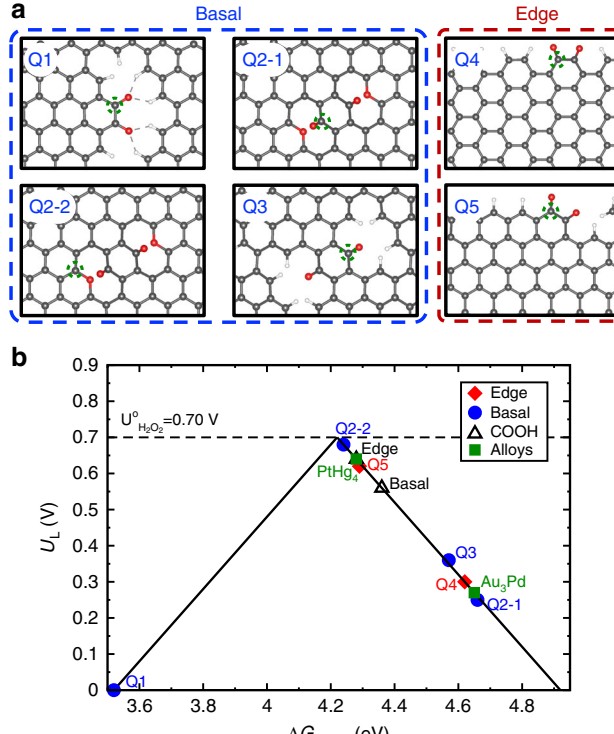

**Fig. 6 Theoretical analysis of different oxygenated groups. a** The atomic structures of the examined oxygen functional groups. Color code: carbon, gray; oxygen, red; hydrogen, white. The corresponding examined active sites are marked with a dashed green circle in each model structure. **b** Theoretical ORHP activity volcano plot. Horizontal dashed line corresponds to the thermodynamic equilibrium potential for ORHP ($U^0 = 0.70$ V). The activity of alloys and edge COOH are adapted from refs. [1,16], respectively.

(Eq. 1). The second electron transfer results in the formation of $HO_2^-$, which is desorbed from the surface (Eq. 2). The key intermediate OOH* plays a pivotal role in the ORHP. Its adsorption in Eq. 1 and desorption in Eq. 2 jointly determines the activity, according to the Sabatier principle[1,16]. The adsorption energy of OOH* ($\Delta G_{OOH*}$) is therefore the best descriptor to capture the trends in activity for different oxygen functional groups[1,16].

We also calculated the barrier of proton transfer to the oxygen atom of adsorbed OOH* to be zero. This means that there is a close connection between the thermodynamic and kinetic formulations for two-electron oxygen reduction reaction. Therefore, we only focus on thermodynamic analysis, which has played an essential role in providing insights on the nature of active sites and guiding the design and optimization of various catalysts.

In line with previous reports[1,16], we used the calculated limiting potential ($U_L$) as the indicator of activity towards ORHP, which is defined as the maximum potential at which the above two reaction steps are downhill in free energy. The ORHP results in this work on quinone functional groups are summarized along with the previous report of oxygen functional groups by Lu, et al.[16] in an activity volcano plot in Fig. 6b. The vertex of the activity volcano corresponds to the thermodynamic equilibrium potential ($U^0 = 0.70$ V) for the ORHP. In theory, an ideal catalyst should have a $\Delta G_{OOH*}$ of 4.22 ($\pm 0.1$) eV, which provides the highest activity. Based on this analysis, while the quinone functional groups on the edge (Q-edge 5) are comparable to the reported catalysts for ORHP[1,16], the Q-basal 2-2 displayed the highest activity. Of note, the formation of quinone functional

groups on the edge seems more feasible than in the basal area, because the formation of Q-Basal groups significantly interrupts the $sp^2$ network and requires a lot of energy input. However, the edge-located structures of Q-Edge 4 and 5 are easily formed, because the $sp^2$ C–C bond breaking is lower. Thus, the Q-Edges are the most likely sites.

Furthermore, a different mechanism similar to industrial anthraquinone process was considered. The results of the anthraquinone mechanism shown in Supplementary Figs. 21 and 22 reveals that the formation of anthrahydroquinone (AHQ) is uphill by 3.64 eV. In addition, the next step, which is transferring proton from of AHQ to $O_2$ molecule and forming radical OOH, is even more exergonic around 4.93 eV. Both of them are far beyond the energy capacity for the two-electron ORR. Therefore, the anthraquinone mechanism is not a possible competitive pathway for our catalyst system.

## Discussion

In summary, we adopted a pre-activated method to decorate the dangled edges of graphitic materials with targeted groups (ether, carboxyl and quinone). The functional groups were then characterized by a combination of soft XANES, XPS, FTIR and CV. Our results confirmed a new class of quinone-edged groups, which exhibited higher selectivity than previously reported oxygenated groups with similar onset potential. The quinone-enriched samples (GNP$_{C=O,1}$) exhibited a $H_2O_2$ yield ratio of 97.8% at 0.75 V. The results were further verified using standalone molecular chemistry and theoretical analysis. These findings will be beneficial for understanding active sites in ORHP, and will be a guide to designing high ORHP catalysts.

## Methods

**Preparation of graphitic nanoplatelets**. The pre-activated method was used for the preparation of the graphitic nanoplatelets (GNP). We first cleaved graphite using the mechanochemical method. The graphite crushing and exfoliation into nanosized particles were accomplished at the same time. The freshly broken edges are free and reactive. Then, the activated edges were reacted with target molecules, such as $CO_2$ and $O_2$. The groups were edge-enriched with $CO_2$ and $O_2$, and the resulting as-prepared GNPs were designated as GNP$_{C=O}$ and GNP$_{C-O-C}$, respectively.

GNP$_{C=O,1}$: the mechanochemical ball-milling method was employed to activate and saturate the graphite at the same time. The preparation was conducted on a planetary ball-milling device (Pulverisette 6, Fritsch GmbH) in a rotation speed of 500 rotation per minutes (rpm). In brief, graphite (15 g, Alfa Aesar, 100 mesh, 99.9995%, product number: 14735), dry ice (100 g, Hanyu Chemical Inc.), and hardened steel balls with a diameter of 3 mm (500 g) were placed into a ball-mill container (250 mL). Then, the air in the container was completely pumped out by five repeated argon (Ar) charging/discharging cycles for 15 min and ball-milled under reduced pressure for 15 h. Finally, the as-prepared GNP$_{C=O}$ was leached in 1 M aq. $H_2SO_4$ solution for 24 h to completely remove possible contamination from unbound Fe debris, followed by rinsing with ultra-pure water (18.2 MΩ cm, Direct-Q® 3UV, Millipore Corporation) more than 6 times and freeze-drying in *tert*-butyl alcohol. Finally, the samples were further dried in vacuum oven at 80 °C for 10 h.

GNP$_{C=O,2}$: the amount of carbonyl-related group loading was controlled by varying the graphite loading amounts and ball-milling conditions. GNP$_{C=O,2}$ was prepared with more carbonyl-related groups by ball-milling graphite (10 g) and dry ice (100 g) with hardened steel balls (500 g, $\Phi = 5$ mm) for 40 h.

GNP$_{C=O,3}$: the different contents of carbonyl-related groups was obtained by $H_2O_2$ leaching in acid. The leaching process was conducted by immersing GNP$_{C=O,2}$ in 20 mL 3.5 M aq. $H_2O_2$ and 1.0 M acetic acid mixture for 12 h. Here, acetic acid acted as a stabilizer to suppress the spontaneous decomposition of $H_2O_2$. Finally, the samples were further dried in a vacuum oven at 80 °C for 10 h before characterizations.

GNP$_{C-O-C}$: the mechanochemical ball-milling method was first applied for activation. The experiment procedures were conducted in a planetary ball-milling device (Pulverisette 6, Fritsch GmbH) at a rotation speed of 500 rpm with the protection of 5 bar Ar (UHP, 99.999%, N50, KOSEM, Korea). In brief, graphite (15 g) and hardened steel balls (500 g, $\Phi = 3$ mm) were charged in a ball-mill container (250 mL). Then, the container was filled with argon gas (5 bar), after five purging cycles with the aid of a vacuum pump to remove residual air.

After cooling to room temperature, the container was filled with an $O_2$/Ar mixture (10 vol%) for 6 h at a flow rate of 250 standard cubic centimeters per

minute (sccm). SAFETY NOTE: The concentration of $O_2$ in the gas mixture should be lower than the burn-off point to avoid fire, which can be caused by violent oxidation.

Since the dangling edges activated by unzipping the graphitic framework tend to reconstruct spontaneously to reduce their surface energy, the cleavage process was exponentially reduced as the ball-milling was prolonged. Gas oxidation in the $O_2$/Ar mixture was divided into 7 periods. The ball-milling time for each period was 20 min, 20 min. 20 min, 30 min, 30 min, 60 min, and 120 min, respectively. The total ball-milling time was 5 h.

Finally, to completely remove unbound Fe debris, the as-prepared $GNP_{C-O-C}$ was leached in 1 M aq. $H_2SO_4$ solution for 24 h, followed by rinsing with ultra-pure water more than six times and freeze-drying in *tert*-butyl alcohol. Finally, the samples were further dried in a vacuum oven at 80 °C for 10 h.

**Structural characterization**. The microstructures were characterized on a JEM-2100 transmission electron microscope (TEM) at an accelerating voltage of 200 kV, and by field emission scanning electron microscopy (FESEM, Nova NanoSEM, FEI), equipped with energy dispersive spectroscopy (EDS, EDAX, AMETEK). X-Ray diffraction (XRD) patterns were recorded on a D/max2500V (Rigaku, Japan) using Cu-Kα radiation (40 kV, 100 mA, $\lambda = 1.5418$ Å) in a 2θ range of 3°–60° at a scan rate of 4° min$^{-1}$. The specific surface area was analyzed on a Micromeritics ASAP 2504 N by nitrogen adsorption-desorption isotherms using the Brunauer-Emmett-Teller (BET) method. The pore distributions were calculated by the non-local density functional theory (NLDFT) method.

Fourier transform infrared spectra (FTIR) were collected on a Perkin-Elmer Spectrum 100 with a resolution of ~1 cm$^{-1}$, and the samples were tableted with KBr as support. The Raman spectra were characterized on a WITec Alpha300R with a laser wavelength of 532 nm.

The elemental analysis (EA) was conducted using a Flash 2000 CHNS/O Analyzers (Thermo Scientific). All samples were measured at least three times. The element of iron was detected by time-of-flight secondary ion mass spectrometry (TOF-SIMS, TOF.SIMS5, IONTOF GmbH, Germany) with a resolution of ppm). The primary ion species was Bi with a dose of $2.0 \times 10^9$, and the raster area was about $400 \times 400$ μm$^2$. The X-ray photoelectron spectra (XPS) was recorded on a Thermo Fisher XPS spectrometer (K-alpha), which employed monochromatic Al Ka radiation as the X-ray source.

The soft X-ray absorption near edge structure (XANES) experiments were performed at the BL12B-A beamline in the National Synchrotron Radiation Laboratory (NSRL), University of Science and Technology of China (USTC), Hefei, P. R. China.

**Electrochemical measurements**. The electrochemical measurements were conducted in a three-electrode electrochemical cell on a workstation of CompactStat (Ivium Technologies B.V., Netherlands). A graphite rod (Alfa Aesar, Ultra purity, 99.9995 %) and an Ag/AgCl electrode were selected as the counter electrode and reference electrode, respectively. As-prepared $GNP_{C=O,1}$, $GNP_{C=O,2}$, $GNP_{C=O,3}$, $GNP_{C-O-C}$, and $p$RGO inks were drop-cast on glassy carbon (GC, 0.247 cm$^2$) supports to prepare working electrodes.

To obtain reliable and reproducible measurements, the cleanness of the GC supports is particularly important. Before each measurement, the cleanness of blank GC was first checked by scanning CV at a scan rate of 50 mV s$^{-1}$. The current curves should only exhibit the shape of capacitance with a current density less than an order of magnitude below $10^{-7}$ A. If it was not, the GC support was polished with alumina (0.05 μm), and then ultrasonically cleaned in ethanol and ultra-pure water.

**Computational method**. We used the Atomic Simulation Environment (ASE)[54] to handle the simulation and the QUANTUM ESPRESSO[55] program package to perform electronic structure calculations. The electronic wavefunctions were expanded in plane waves up to a cutoff energy of 500 eV, while the electron density is represented on a grid with an energy cutoff of 5000 eV. Core electrons were approximated using ultrasoft pseudopotentials[56]. To describe chemisorption properties on graphene structures, we used the PBE exchange-correlation functional with dispersion correction[57]. Graphene structures were modeled as one layer with a vacuum of 20 Å to decouple the periodic replicas. To model the quinone functional groups in the basal plane, we use a $5 \times 5$ super cell lateral size, and the Brillouin zone was sampled with $(4 \times 4 \times 1)$ Monkhorst-Pack k-points. For the oxygen functional groups in the edge, we used a super cell with a lateral size $5 \times 6$ and the Brillouin zone was sampled with a $(1 \times 4 \times 1)$ Monkhorst-Pack k-points.

## Data availability

The data that support the findings of this study are available from the corresponding author upon reasonable request.

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

## Acknowledgements

This work was supported by the Creative Research Initiative (CRI, 2014R1A3A2069102), BK21 Plus (10Z20130011057), and Science Research Center (SRC, 2016R1A5A1009405) programs through the National Research Foundation (NRF) of Korea. S.S. acknowledges the support from the University of Calgary's Canada First Research Excellence Fund Program, the Global Research Initiative in Sustainable Low Carbon Unconventional Resources.

## Author contributions

J-B.B. conceived the project and oversaw all the research phases. J-B.B. and G-F.H. designed the project. G-F.H. synthesized and characterized the samples. F.L., Z.F., Y.L., and W.Z. measured the soft XANES. S.S. and M.K. conducted the theoretical calculations. S-W.K. and Y.B. built the two-electrode device. S-K.K. and J-P.J. performed BET measurements. Data collection and analysis were conducted by J-B.B., S.S., and G-F.H. All the authors contributed to and commented on this paper.

## Competing interests

The authors declare no competing interests.
