## [Peer Review File · Nature Communications]

Reviewers' Comments:

Reviewer #1:

Remarks to the Author:

The work by Han et al explores the preparation of a ORR selective metal-free catalyst for the production of H₂O₂, by a rational design based on the well known anthraquinone process, where the quinone units are expected to trigger selective formation of H₂O₂ by oxidized graphitic carbon materials. I like the logical approach and the attempt to determine the several features of the catalysis also from a computational approach, as well as the identification of the active sites. There are however a few issues that in my opinion the authors should address.

Above all, I do not understand the rationale behind evaluating H₂O₂ formation at basic pH (0.1 M KOH). Why acidic pH was not explored? H₂O₂ formation in acidic media is industrially more relevant for several applications as compared to alkaline media. Moreover, neutral pH electrolytes would be compatible with uses of H₂O₂ solutions as cheap disinfectants. In this context, testing activity in strongly basic pH is to me something that diminishes the impact of the work. Moreover, from a catalytic point of view, it is not interesting to evaluate selectivity of the process in alkaline environment, as it has been reported that simple glassy carbon electrodes can lead to a total H₂O₂ selectivity at potential as positive as 0.72V (see ACS Catal. 8, 4064-4081 (2018))

I do not quite understanding how the N doping is introduced in the pRGO reference catalyst.

Moreover, N-doped carbons do not necessarily favour the 4-electron pathway, as it has recently been reported, but this depends on the N type distribution. If the authors presume that N atoms are playing a role in diminishing the selectivity by pRGO, so a more detailed analysis should be done on this material, reporting the % and type the as-introduced N atoms, and discussion expanded.

Porosity of carbon materials also appears to be an important aspect in ORR, so the textural properties of the various materials should be analysed and discussed.

Reviewer #2:

Remarks to the Author:

The manuscript "Building and identifying highly active oxygen functional groups in carbon materials for oxygen reduction to hydrogen peroxide," authored by G.-F. Han, F. Li, W. Zou, J.-P. Jeon, S.-W. Kim, S.-J. Kim, Y. Bu, Z. Fu, Y. Lu, S. Siahrostami, and J.-B. Baek, is an original work involving new results concerning a pre-activated method to decorate the dangled edges of graphitic materials with target groups (ether, carboxyl, and quinone). The authors demonstrated the activity of the quinone functional groups toward the ORR by examining standalone quinone molecular systems, which revealed that a sample containing abundant quinone functional groups (GNPC=O,1) exhibited high selectivity, with a H₂O₂ yield ratio of 97.8%. The study is suitable for the Journal, since the work contains technically sound data, and the paper provides strong evidence for the conclusions drawn. The quality of the presentation is suitable. Also, the manuscript is important to scientists in the specific and wider field.

Points:

Page 22, caption to Figure 3: Authors should state very clearly in the caption the rotation rate (rpm) for the results shown were obtained. The same approach should be applied to the entire manuscript regarding RRDE and RDE responses.

Page S6: Authors should state very clearly in the Supp. Inf. the geometric area of the GC electrode, as well as the loading (at $\mu\text{g cm}^{-2}$) of catalysts used at the GC surface.

Page S8: Authors should state very clearly in the Supp. Inf. the geometric area of the disc and ring electrodes (RRDE) as well as the material (composition) of the disc and ring electrodes (RRDE).

Gilberto Maia

Reviewer #3:

Remarks to the Author:

In this work, the authors used experiments combined with theory to identify the active sites on oxygen-functionalized graphene for electrochemical oxygen reduction to hydrogen peroxide (ORHP).

They concluded that the quinone-edged group is the active center.

It is my opinion that the theoretical part is a problem to the whole manuscript. Instead of investigating the detail reaction mechanism of how O₂ is reduced to H₂O₂ by proton-electron pairs, the authors simply used $\Delta G(*OOH)$ as a descriptor. However, the descriptor approach is effective only when the underlying reaction mechanism is the same between the target system and the systems that the approach is built upon. The $\Delta G(*OOH)$ descriptor is developed for the metallic electrodes, in which ORHP involves two steps: (a) $* + O_2 \rightarrow *OOH$, and (b) $*OOH + H^+ + e^- \rightarrow H_2O_2$ (ref: J. Am. Chem. Soc. 2018, 140, 7851). This mechanism is unlikely to be suitable for oxygen-functionalized graphenes. Oxygen-functionalized graphene is more likely to catalyze ORHP in the same fashion as the industrial anthraquinone process, in which anthraquinone is first hydrogenated to anthrahydroquinone, and then oxygen is reduced by anthrahydroquinone to produce H₂O₂ and recover anthraquinone (ref: Eur. J. Org. Chem. 2011, 4113). Based on this, it is my opinion that this work is not suitable for Nat. Commun.

Response to the Reviewers

Jong-Beom Baek (On the behalf of all authors, who contributed to this work).

As a whole, the reviewers' comments were very helpful to improve the quality of the manuscript. We are very grateful to the reviewers for their time and valuable comments.

Reviewer #1: (Remarks to the authors)

The work by Han et al explores the preparation of a ORR selective metal-free catalyst for the production of H₂O₂, by a rational design based on the well known anthraquinone process, where the quinone units are expected to trigger selective formation of H₂O₂ by oxidized graphitic carbon materials. I like the logical approach and the attempt to determine the several features of the catalysis also from a computational approach, as well as the identification of the active sites.

Response #1: We thank the reviewer #1 for his/her highly positive and encouraging comments. We also appreciate the reviewer for the constructive suggestions. Following the reviewer's suggestions and comments, we have revised the manuscript. The details are listed below.

There are however a few issues that in my opinion the authors should address. Above all, I do not understand the rationale behind evaluating H₂O₂ formation at basic pH (0.1 M KOH). Why acidic pH was not explored? H₂O₂ formation in acidic media is industrially more relevant for several applications as compared to alkaline media.

Response 1.1: Thanks for the reviewer's comment. In the view of industry, **H₂O₂ in the alkaline media is more important than in acidic ones**. We can draw the conclusion from its end-user applications. The **Fig. R1** shows the global H₂O₂ market by end user, according to the report by Technavio. As we can see, most of H₂O₂ are used for **bleaching-related fields**, namely textiles (**16 %**) as well as pulp and paper (**48%**). In those bleaching fields, the **exclusive selection is alkaline media** because of the following reasons:

- a) During the bleaching, the **active substance is the hydroxyl radical (OH^{*})**, which is formed by decomposition of H₂O₂. Obviously, the **bleaching effect** of H₂O₂ in alkaline media is **much more efficient** than that in acidic media due to the **high amount of OH^{*} in base**¹. That is to say that the bleaching effect of H₂O₂ will be discounted in the neutral and especially acidic media.
- b) Bleaching in alkaline condition also has a cleaning effect. Therefore, there are combining advantages in both alkaline extraction and bleaching treatment.

It means that **as least 64%** of H_2O_2 are used in alkaline media. Thus, the alkaline media are more important than the acidic ones.

Fig. R1. Global Hydrogen Peroxide Market 2017-2021: Industry Insights, Segmentation, and Forecasts by Technavio.

<https://www.businesswire.com/news/home/20170724006216/en/Global-Hydrogen-Peroxide-Market-2017-2021-Industry-Insights>

The above is based on the consideration of end user. Next, we will discuss this issues in the view of electro-synthesizing H_2O_2 **in reality**. **The alkaline media are also superior to the acidic ones** due to the following reasons:

The electrode stability is a big issue in acidic conditions, since acid is more corrosive than alkaline. During designing the electrochemical device, both cathode and anode should be considered. Although it is not a big problem for the cathode, which reduces the O_2 into H_2O_2 , it is a very challenging problem for the anode, which occurs the oxidation of water into O_2 , namely oxygen evolution reaction (OER). Usually, the applied potential on the anode should be higher than 1.5 V. There is nearly no catalyst, which shows long-term stability at such high potential. Thus, **H_2O_2 electrochemical synthesis is better realized in the alkaline media.**

In summary, **the importance of H_2O_2 synthesis in alkaline media is much more important than that in acidic media in consideration of both end user and economic feasibility in practice.**

In response to this specific suggestion, **we have added the measurements in acidic media**, and discussed in the revised main text. The data is shown in **Fig. R2a,b** and **Supplementary Fig. 15** in Supporting Information. Our data agree well with the previous reports, that the oxygenated carbon materials show poor activity in the acidic media²⁻⁵. However, **the quinone-enriched samples ($\text{GNP}_{\text{C=O},1}$) still exhibits the highest activity, approximately 85%**. Here, it is less meaningful to discuss about the exact onset potential of ORHP, because the non-negligible $J_{\text{H}_2\text{O}}$ impedes the exact extraction of onset potential. The rough overpotential is about 0.3 V relative to theoretical equilibrium potential.

Fig. R2. a, The polarization curves of H₂O₂ current (J_R or $J_{H_2O_2}$), disk current (J_D), and H₂O current (J_{H_2O}). The curves were measured in O₂-saturated 0.1 M HClO₄ solution at a scan rate of 10 mV s⁻¹ by RRDE with a rotation speed of 1600 rpm. The applied potential of the ring was 1.15 V. The current was the average of the forward and backward scans. **b,** The corresponding H₂O₂ yield ratio. **c,** The polarization curves were measured in O₂-saturated 0.05 M Na₂SO₄ solution at a scan rate of 10 mV s⁻¹ by RRDE with a rotation speed of 1600 rpm. The applied potential of the ring was 1.15 V. The current was the average of the forward and backward scans. **d,** The corresponding H₂O₂ yield ratio.

Moreover, neutral pH electrolytes would be compatible with uses of H₂O₂ solutions as cheap disinfectants. In this context, testing activity in strongly basic pH is to me something that diminishes the impact of the work.

Response 1.2: Thanks for the reviewer's constructive suggestion. We agree that neutral pH electrolytes would be compatible with uses of H₂O₂ solutions as cheap disinfectants. Following the reviewer #1's suggestion, we conducted the experiment in a neutral media (0.05 M Na₂SO₄). Our results determined that the quinone-enriched samples (GNP_{C=0,1}) are also good ORHP candidate in neutral media.

Although phosphate buffered saline (PBS) is a frequently used neutral media, which is used in some literatures for oxygen reduction to hydrogen peroxide (ORHP), our experiments determined that the **Pt ring in the RRDE will be dissolved while**

detecting H_2O_2 at the applied potential of 1.15 V (Fig. R3) due to the presence of Cl^- . That is why we did not select PBS as test solution.

Fig. R3. Cyclic voltammetry, which was measured in Ar-saturated commercial phosphate buffered saline (PBS) solution at a scan rate of 50 mV s^{-1} . The Pt dissolution at high potential indicates that the Cl^- -containing PBS solution cannot be used as test solution during RRDE measurement.

As shown in **Fig. R2c,d** and **Supplementary Fig. 15**, the performance of oxygen reduction to hydrogen peroxide (ORHP) was evaluated in $0.05 \text{ M Na}_2\text{SO}_4$ solution. In comparison with the obtained data in the alkaline media, the most obvious difference is the absence of potential shift, which is caused by the local pH change. The onset potential and H_2O_2 yield ratio follow the same trends in alkaline condition. **The quinone-enriched sample ($\text{GNP}_{\text{C}=0.1}$) also exhibits the highest selectivity in neutral media.**

*Moreover, from a catalytic point of view, it is not interesting to evaluate selectivity of the process in alkaline environment, as it has been reported that simple glassy carbon electrodes can lead to a total H_2O_2 selectivity at potential as positive as 0.72V (see *ACS Catal.* 8, 4064-4081 (2018))*

Response 1.3: Thanks for the reviewer's thoughtful comment. Yes, our experiments also verify the reported result (*ACS Catal.* 8, 4064-4081 (2018)), which glassy carbon can lead to a nearly total H_2O_2 selectivity (**Fig. R4** and **Fig. 5 in Main Text**). However, its limiting current and onset potential are too low. That is to say, **its activity is very poor**. Thus, it **cannot be used for the electrochemical synthesis of H_2O_2 in reality**.

The extended glassy carbon and graphitic carbon **have different chemical nature**. The conclusion drawn by no one can be simply transplanted to another one. The active sites on the two kinds of carbon materials are different, which can be proven by the fact that the active sites on glassy carbon have high selectivity but poor activity. Our sample possesses both high activity and selectivity. The result implies that **the active sites are different between the two carbon materials**.

Fig. R4. The comparison between blank glassy carbon (GC) and GNP_{C=0.1}. The polarization curves of H₂O₂ current (J_R or $J_{H_2O_2}$), disk current (J_D), and H₂O current (J_{H_2O}). The curves were measured in O₂-saturated 0.1 M KOH solution at a scan rate of 10 mV s⁻¹ by RRDE. The applied potential of the ring was 1.15 V. The current was the average of the forward and backward scans.

I do not quite understanding how the N doping is introduced in the pRGO reference catalyst.

Response 1.4: Thanks for the reviewer's comment. We adopted a commonly used method to prepare the reduced graphene oxide (RGO), following the work by Prof. Wallace [*Nat. Nanotechnol.* **3**, 101–105 (2008)]⁶. During the reduction of graphene oxide (GO), **the use of hydrazine monohydrate (NH₂NH₂ · H₂O) will inevitably introduce the N moieties into the RGO**. The content of N is approximately 6.4 wt.% (determined by elemental analysis, see the annotation c in **Supplementary Table 1**). Because we did not use other methods to further reduce the remnant N and O in our RGO sample, that is why we denotes our RGO sample as partially reduced graphene oxide (*pRGO*).

Moreover, N-doped carbons do not necessarily favour the 4-electron pathway, as it has recently been reported, but this depends on the N type distribution. If the authors presume that N atoms are playing a role in diminishing the selectivity by pRGO, so a more detailed analysis should be done on this material, reporting the % and type the as-introduced N atoms, and discussion expanded.

Response 1.5: We agree with the reviewer's comment that the activity of N-doped carbon depends on the N type.

According to the suggestion, we measured the XPS of pRGO (**Fig. R5** and **Supplementary Fig. 14**). Our experimental results indicated that the N 1s composed of amino N (399.2 eV, 34%), pyrrolic N (400.1 eV, 43%) and graphitic N (401.4 eV, 23%). The graphitic N is thought to be active sites for 4e⁻ ORR⁷. Moreover, a discussion is added in the revised main text with highlight.

Fig. R5. The high-resolution N 1s of the X-ray photoelectron spectra (XPS) for pRGO. Amino N, 399.2 eV, 34%; Pyrrolic N, 400.1 eV, 43%; Graphitic N, 401.4 eV, 23%.

Porosity of carbon materials also appears to be an important aspect in ORR, so the textural properties of the various materials should be analysed and discussed.

Response 1.6: Thanks for the reviewer's thoughtful comment. We conducted additional BET measurements to test textural properties of the various materials (**Fig. R6** and **Supplementary Fig. 4**). The specific BET areas of GNP_{C=0,1}, GNP_{C=0,2}, GNP_{C=0,3}, and GNP_{C=0,C} are 450, 753, 657, and 401 m² g⁻¹, respectively. The insets show the pore size distributions (PSD). Since our samples have typical morphologies

of nanoplatelet or nanoparticle. The two pores at approximately 1 nm and higher than 1 nm are attributed to slit- and cylindrical-pores between nanoplatelets or nanoparticles.

Fig. R6. The Brunauer–Emmett–Teller (BET) measurements of $\text{GNP}_{\text{C}=0,1}$, $\text{GNP}_{\text{C}=0,2}$, $\text{GNP}_{\text{C=0-C}}$, and $\text{GNP}_{\text{C}=0,3}$. The insets are the pore size distributions (PSDs), which are analyzed using non-local density functional theory (NLDFT) model.

Reviewer #2: (Remarks to the authors)

The manuscript “Building and identifying highly active oxygen functional groups in carbon materials for oxygen reduction to hydrogen peroxide,” authored by G.-F. Han, F. Li, W. Zou, J.-P. Jeon, S.-W. Kim, S.-J. Kim, Y. Bu, Z. Fu, Y. Lu, S. Siahrostami, and J.-B. Baek, is an original work involving new results concerning a pre-activated method to decorate the dangled edges of graphitic materials with target groups (ether, carboxyl, and quinone). The authors demonstrated the activity of the quinone functional groups toward the ORR by examining standalone quinone molecular systems, which revealed that a sample containing abundant quinone functional groups ($\text{GNP}_{\text{C=O},1}$) exhibited high selectivity, with a H_2O_2 yield ratio of 97.8%. The study is suitable for the Journal, since the work contains technically sound data, and the paper provides strong evidence for the conclusions drawn. The quality of the

presentation is suitable. Also, the manuscript is important to scientists in the specific and wider field.

Response #2: We thank the reviewer #2 for his/her highly positive and constructive comments. We also appreciate very comprehensive suggestions to further improve the quality of the work.

Points:

Page 22, caption to Figure 3: Authors should state very clearly in the caption the rotation rate (rpm) for the results shown were obtained. The same approach should be applied to the entire manuscript regarding RRDE and RDE responses.

Response 2.1: We thank the reviewer for carefully reading of our work. We have stated the rotation speed (1600 rpm) in all the LSV curves.

Page S6: Authors should state very clearly in the Supp. Inf. the geometric area of the GC electrode, as well as the loading (at g cm²) of catalysts used at the GC surface.

Response 2.2: Thanks for the thoughtful suggestion. We have clearly stated the geometric area of GC electrode (0.247 cm²). The loadings of the catalysts were also given in the Supporting Information. The loadings of GNP_{C=0,1}, GNP_{C=0,2}, GNP_{C=0,3}, GNP_{C-O-C}, and pRGO were approximately 0.29, 0.22, 0.60, 0.25 and 0.12 mg cm⁻², respectively.

Page S8: Authors should state very clearly in the Supp. Inf. the geometric area of the disc and ring electrodes (RRDE) as well as the material (composition) of the disc and ring electrodes (RRDE).

Response 2.3: We thank the reviewer for the specific comment. According to the suggestion, the following sentences were added in the Supporting Information.

"The disk electrode was made of glassy carbon with a diameter of 5.61 mm (area, 0.247 cm²). The Pt ring electrode had an outer diameter of 7.92 mm and an inner diameter of 6.25 mm with a ring-disk gap of 320 μm (Model: AFE7R9GCPT)."

Reviewer #3: (Remarks to the authors)

In this work, the authors used experiments combined with theory to identify the active sites on oxygen-functionalized graphene for electrochemical oxygen reduction to hydrogen peroxide (ORHP). They concluded that the quinone-edged group is the active center.

Response #3: We thank the reviewer #3 for his/her opinion on our work. We also appreciate the reviewer's approval on our experimental section.

*It is my opinion that the theoretical part is a problem to the whole manuscript. Instead of investigating the detail reaction mechanism of how O₂ is reduced to H₂O₂ by proton-electron pairs, the authors simply used deltaG(*OOH) as a descriptor. However, the descriptor approach is effective only when the underlying reaction mechanism is the same between the target system and the systems that the approach is built upon. The deltaG(*OOH) descriptor is developed for the metallic electrodes, in which ORHP involves two steps: (a) * + O₂ → *OOH, and (b) *OOH + H₂O₂ (ref: J. Am. Chem. Soc. 2018, 140, 7851). This mechanism is unlikely to be suitable for oxygen-functionalized graphenes.*

Response 3.1: Thanks for the reviewer's comments. The cited pathway by the reviewer #3 (ref: J. Am. Chem. Soc. 2018, 140, 7851) is a typical reaction mechanism in acid, as shown in Equation 1:

In present works, we mainly focus on the base environment. Its pathway is shown in Equation 2:

As underlined above, the **only involved intermediate** in both acid and base environment is **OOH***, which is the **key** intermediate in both acid and base mechanisms. The **OOH*** is formed by adsorbing O₂ at the surface and taking proton from solution. The proton source in **acid** environment is **hydronium**, while the proton source in **base** is **water**. **That is the only difference between acid and base conditions.**

Adsorption energy is a **surface property**, which is **not affected by the proton source**. In DFT calculations, we model the reaction mechanism by taking into account the adsorption energies of intermediates of each mechanism. We have proven in the past that adsorption energy of **OOH*** is the **best descriptor of activity in both acid and base mechanisms**. That is why we model everything based on RHE scale. This led us to this conclusion that is in reality. The ability of the surface-active sites toward OOH* intermediate is suitable for determining the activity under acid and base conditions. The conclusion was further verified by the fact that the quinone-enriched samples showed the highest selectivity in alkaline, neutral, and acidic media.

Oxygen-functionalized graphene is more likely to catalyze ORHP in the same fashion as the industrial anthraquinone process, in which anthraquinone is first hydrogenated to anthrahydroquinone, and then oxygen is reduced by anthrahydroquinone to produce H₂O₂ and recover anthraquinone (ref: Eur. J. Org. Chem. 2011, 4113). Based on this, it is my opinion that this work is not suitable for Nat. Commun.

Response 3.2: Thanks for the reviewer's comments. Our process is a **catalytic process**, not the industrial anthraquinone process, which is announced by the reviewer #3. It can be proved by the following facts:

- (1) Our process is a **continuously time-independent** process, which is a **typical character of catalytic process**. Differently, the industrial anthraquinone process is **time-dependent**. The reaction rate will be **reduced** due to the **depletion of anthrahydroquinone** as the reaction is proceeding. As shown in **Fig. R7**, the current can even be maintained stable as long as 120 h. If our process is an industrial anthraquinone process, the current should be drop down to zero. Evidently, **our process a catalytic process**.
- (2) The **onset potential of our process** is **higher (130 mV)** than the **onset potential of the reduction of anthrahydroquinone to anthraquinone (Fig. R8)**. In the view of thermodynamics, our process is superb to an industrial anthraquinone process.
- (3) Since industrial anthraquinone process is a redox reaction, which contains both reduction and oxidation processes, if our process is an industrial anthraquinone process, there should be an oxidation current during CV scanning. However, **our process only contains reduction current (Fig. R9)**. It further demonstrates that our process is **not** an industrial anthraquinone process.

Fig. R7. The current-time curves in four fresh 0.1 M KOH electrolytes, which were measured in a H-type cell, with 400 ppm MgSO₄ as stabilizer to suppress the decomposition of H₂O₂. The applied potential was 0.65 V. The current exhibited a slow decline as the time increased. The possible reason is that the increasing H₂O₂

concentration enhances electrolyte viscosity. The increasingly sluggish diffusion of O_2 in the H_2O_2 solution subsequently deteriorates the ORHP current⁸.

Fig. R8. **a**, The polarization curves of $J_{H_2O_2}$, J_D , and J_{H_2O} . The curves were measured in O_2 -saturated 0.1 M KOH solution at a scan rate of 10 mV s^{-1} by RRDE. The current was the average of the forward and backward scan. **b**, Cyclic voltammetry, which was measured in Ar-saturated 0.5 M H_2SO_4 at a scan rate of 50 mV s^{-1} . The sample is $GNP_{C=0,2}$.

Fig. R9. The polarization curves of disk current with positive and negative scan. The curves were measured in O_2 -saturated 0.05 M Na_2SO_4 solution at a scan rate of 10 mV s^{-1} by RRDE. The sample is $GNP_{C=0,1}$.

References

1. Ribeiro, A. R., Nunes, O. C., Pereira, M. F. R. & Silva, A. M. T. An overview on the advanced oxidation processes applied for the treatment of water pollutants defined in the recently launched Directive 2013/39/EU. *Environ. Int.* **75**, 33–51 (2015).
2. Kim, H. W. *et al.* Efficient hydrogen peroxide generation using reduced graphene oxide-based oxygen reduction electrocatalysts. *Nat. Catal.* **1**, 282–290 (2018).
3. Lu, Z. *et al.* High-efficiency oxygen reduction to hydrogen peroxide catalysed by oxidized carbon materials. *Nat. Catal.* **1**, 156–162 (2018).
4. Sa, Y. J., Kim, J. H. & Joo, S. H. Active Edge-Site-Rich Carbon Nanocatalysts with Enhanced Electron Transfer for Efficient Electrochemical Hydrogen Peroxide Production. *Angew. Chemie Int. Ed.* **58**, 1100–1105 (2019).
5. Chen, Z. *et al.* Development of a reactor with carbon catalysts for modular-scale, low-cost electrochemical generation of H₂O₂. *React. Chem. Eng.* **2**, 239–245 (2017).
6. Li, D., Müller, M. B., Gilje, S., Kaner, R. B. & Wallace, G. G. Processable aqueous dispersions of graphene nanosheets. *Nat. Nanotechnol.* **3**, 101–105 (2008).
7. Dai, L., Xue, Y., Qu, L., Choi, H.-J. & Baek, J.-B. Metal-Free Catalysts for Oxygen Reduction Reaction. *Chem. Rev.* **115**, 4823–4892 (2015).
8. Ruiz-Ibanez, G., Bidarian, A., Davis, R. A. & Sandall, O. C. Solubility and diffusivity of oxygen and chlorine in aqueous hydrogen peroxide solutions. *J. Chem. Eng. Data* **36**, 459–466 (1991).

Reviewers' Comments:

Reviewer #1:

Remarks to the Author:

I think the authors have addressed carefully all my issues, and answered in a satisfying manner, added a large load of additional experiments, and therefore I do not have objections in publication in Nature Communications. I just want to make a comment to the authors: apart from industrial applications, that surely USE the H₂O₂ at basic pH, the STORAGE of H₂O₂ is better pursued in acidic solutions, given the higher instability of H₂O₂ in alkaline media. Thus, electro-synthesis in acidic media could directly produce the storable solution (that can be diluted if necessary), that is currently what several industries are attempting to achieve via electrocatalytic methods, to the best of my knowledge.

Reviewer #3:

Remarks to the Author:

I appreciate the authors performed additional experiments. However, I still have serious doubts about the computational part of this paper listed as follows:

(1) The authors only calculated the thermodynamics and completely ignored the kinetics of the proposed ORR mechanism. This could cause serious problems since favorable thermodynamics does not guarantee facile kinetics. I suggest the authors calculate the kinetic barriers at least for the most reactive site that they discovered.

(2) In my previous comments, I mentioned that the ORR to H₂O₂ could go through a mechanism similar to the industrial anthraquinone process, in which anthraquinone is converted to anthrahydroquinone by adding two proton-electron pairs. This is still a catalytic process. The authors can simply perform DFT calculations to investigate this mechanism, and by comparing the free energy surface to those based on their proposed mechanism to provide more solid evidence.

Response to the Reviewers

Jong-Beom Baek (On the behalf of all authors, who contributed to this work).

As a whole, the reviewers' comments were very helpful to improve the quality of the manuscript. We are very grateful to the reviewers for their time and valuable comments.

Reviewer #1: (Remarks to the authors)

I think the authors have addressed carefully all my issues, and answered in a satisfying manner, added a large load of additional experiments, and therefore I do not have objections in publication in Nature Communications. I just want to make a comment to the authors: apart from industrial applications, that surely USE the H_2O_2 at basic pH, the STORAGE of H_2O_2 is better pursued in acidic solutions, given the higher instability of H_2O_2 in alkaline media. Thus, electro-synthesis in acidic media could directly produce the storable solution (that can be diluted if necessary), that is currently what several industries are attempting to achieve via electrocatalytic methods, to the best of my knowledge.

Response #1: We thank the reviewer #1 for his/her highly positive and encouraging comments. We fully agree with the reviewer #1's additional comments.

Reviewer #3: (Remarks to the authors)

I appreciate the authors performed additional experiments. However, I still have serious doubts about the computational part of this paper listed as follows:

Comment 3.1: *The authors only calculated the thermodynamics and completely ignored the kinetics of the proposed ORR mechanism. This could cause serious problems since favorable thermodynamics does not guarantee facile kinetics. I suggest the authors calculate the kinetic barriers at least for the most reactive site that they discovered.*

Response 3.1: We appreciate the reviewer #3 for making this suggestion. To address the reviewer comments, we calculated the transition state energy for transferring a proton from a hydronium ion in the solution to the oxygen atom of adsorbed OOH* on our most stable structure using nudge elastic band (NEB) calculations. Our results show that the proton transfer to the OOH* is completely downhill with zero barrier. This phenomenon has been observed previously for other systems such as transition metals and reported in several publications, including Hansen et al, *J. Phys. Chem. C* 2014, 118, 6706. As a result of zero barrier for proton transfer to oxygenated intermediates, predicted thermodynamic activity volcano has a close agreement with the predicted kinetic activity volcano. Therefore, there is a close connection between the thermodynamic and kinetic formulations for two-electron oxygen reduction reaction. In addition, since we are looking at the trends in activity between different possible active sites, our analysis is valid with both thermodynamic and kinetic analyses.

To address the reviewer's comment, we added the following statement in our computational details in Main Text on page 14.

“We also calculated the barrier of proton transfer to the oxygen atom of adsorbed OOH* to be zero. This means that there is a close connection between the thermodynamic and kinetic formulations for two-electron oxygen reduction reaction. Therefore, we only focus on thermodynamic analysis, which has played an essential role in providing insights on the nature of active sites and guiding the design and optimization of various catalysts.”

Comment 3.2: *In my previous comments, I mentioned that the ORR to H₂O₂ could go through a mechanism similar to the industrial anthraquinone process, in which anthraquinone is converted to anthrahydroquinone by adding two proto-electron pairs. This is still a catalytic process. The authors can simply perform DFT calculations to investigate this mechanism, and by comparing the free energy surface to those based on their proposed mechanism to provide more solid evidence.*

Response 3.2: We thank the reviewer for this comment. To address this point, we calculated the free energy diagram for the mechanism suggested by the reviewer for one of our sample quinone structures (**Figure R1** and **Supplementary Fig. 21**). The standard potential for running the two-electron oxygen reduction reaction is only 0.7 V. And also, because it is a two-electron process, the maximum free energy we can expect is 1.4 eV for the whole reaction. The results of anthraquinone mechanism shown in **Figure R1** below reveals that the formation of anthrahydroquinone (AHQ) is uphill by 3.64 eV. This is by far beyond the energy capacity for the two-electron ORR and by no means is a compatible pathway with the pathway we have presented in our work. In addition, the next step, which is transferring proton from of AHQ to O₂ molecule and forming radical OOH, is even more exergonic around 4.93 eV. Therefore, we conclude that the anthraquinone mechanism is not a possible competitive pathway for our catalyst system, supporting that our previous analysis is valid.

To address the reviewer's comment, we added the above discussion in Main Text on page 15 and **Supplementary Fig. 21** in the SI on page S32.

“Furthermore, a different mechanism similar to industrial anthraquinone process was considered. The results of the anthraquinone mechanism shown in Supplementary Fig. 21 reveals that the formation of anthrahydroquinone (AHQ) is uphill by 3.64 eV. In addition, the next step, which is transferring proton from of AHQ to O₂ molecule and forming radical OOH, is even more exergonic around 4.93 eV. Both of them are far beyond the energy capacity for the two-electron ORR. Therefore, the anthraquinone mechanism is not a possible competitive pathway for our catalyst system.”

Figure R1. Calculated reaction free energy diagram for a mechanism, which is similar to the industrial anthraquinone process. AQ and AHQ denote anthraquinone and anthrahydroquinone, respectively.

Reviewers' Comments:

Reviewer #3:

Remarks to the Author:

I appreciate the authors' effort to answer my questions. I still have some comments about the computational part. In the revised manuscript, the authors found that proton transfer from a hydronium ion in the solution to the oxygen atom of adsorbed OOH* has no barrier. The authors should detail what model they used for this hydronium ion in the manuscript or SI. Also, the transition state should be re-calculated using either CI-NEB or Dimer method, as it has been found in numerous theoretical studies that the NEB method very often underestimates the barriers.

Reviewer #3 (Remarks to the Author):

I appreciate the authors' effort to answer my questions. I still have some comments about the computational part. In the revised manuscript, the authors found that proton transfer from a hydronium ion in the solution to the oxygen atom of adsorbed OOH has no barrier. The authors should detail what model they used for this hydronium ion in the manuscript or SI. Also, the transition state should be re-calculated using either CI-NEB or Dimer method, as it has been found in numerous theoretical studies that the NEB method very often underestimates the barriers.*

Response:

We thank the reviewer for this comment and would like to emphasize that **modeling coupled proton-electron transfer to the adsorbate is challenging and beyond the scope of this work**. However, per the reviewer's request we have provided some details about our calculations here, though **we prefer not to report any of these results in the manuscript as they are trivial and the results have no physical meaning**.

Figure R1a and b display the initial and final structures, respectively, for calculating proton transfer from hydronium ion to the adsorbed OOH* (circled in blue). In order to be able to calculate the proton transfer barrier, we need to model the solvent (monolayer of water structure) near the surface. **Finding a suitable water structure is a difficult task and requires exploring a large ensemble of different water structures to identify the minimum structure**. In a previous report on metal-doped graphene system (*ACS Cent. Sci.* 2017, 3, 12, 1286-1293), we spent months to identify the minimum water structure using minima hopping method (*J. Chem. Phys.* 2004, 120 (21), 9911) near the graphene structure. The optimized water structure found from that study showed hexagonal ice-like water network, which was in very good agreement with experimental observation for water structure on graphene (*J. Am. Chem. Soc.* 2009, 131 (35), 12838). To model coupled proton–electron transfers, we used this monolayer water structure and added an excess hydrogen (yellow sphere, Figure R1a), **resulting in a spontaneous separation of charge at the interface where the excess hydrogen atoms transferred electrons to the surface** (*Chem. Phys. Lett.* 2008, 466, 68). The entire system is therefore electroneutral, and no compensating homogeneous background charge is added.

We observed that spontaneous transfer of proton to OOH* occurs, which is because driving force for the proton transfer is too strong removing any barrier. To be able to do CI-NEB calculations per the reviewer's request, we had to freeze the water structure. **The results, unfortunately, have no physical meaning and cannot be used to draw any meaningful conclusion**. However, for the interest of the reviewer, **the CI-NEB calculations on this artificial system showed 0.4 eV for the activation barrier**.

In general, small barriers for proton transfer to oxygenated intermediates is a known phenomenon in the field that results in a close agreement between thermodynamically predicted activity volcano and predicted kinetic activity volcano (Hansen et al, *J. Phys. Chem. C* 2014, 118, 6706). Therefore, there is a close connection between the thermodynamic and kinetic formulations for two-electron oxygen reduction reaction. We also would like to emphasize again that **our study herein considers the trends in activity between different possible active sites**.

Therefore, even slightly higher or lower barrier in proton transfer to OOH* will not affect the results of our analysis, because this change will apply similarly for all the examined active sites.

Figure R1: **a**, Initial structure with OOH* adsorbed on graphene surface (circled in blue) and proton in the water structure. **b**, Final structure showing the proton transfer to OOH* adsorbate and forming H₂O₂ product (circled in blue).